# A complex structure of escaping helium spanning more than half the orbit of the ultra-hot Jupiter WASP-121 b

Romain Allart [1] ✉, Louis-Philippe Coulombe [1], Yann Carteret [2], Jared Splinter[3,4], Lisa Dang [1,5], Vincent Bourrier [2], David Lafrenière [1], Loïc Albert [1], Étienne Artigau [1], Björn Benneke [1,6], Nicolas B. Cowan [3,4], René Doyon [1], Vigneshwaran Krishnamurthy [3], Ray Jayawardhana[7], Doug Johnstone [8,9], Adam B. Langeveld [7,10], Michael R. Meyer [11], Stefan Pelletier [2], Caroline Piaulet-Ghorayeb[12], Michael Radica [12], Jake Taylor[13] & Jake D. Turner [10]

Atmospheric escape of close-in exoplanets, driven by stellar irradiation, influences their evolution, composition, and atmospheric dynamics. The near-infrared metastable helium triplet (10833 Å) has become a key probe of this process, enabling mass loss rate measurements for dozens of exoplanets. Only a few studies, however, have detected absorption beyond transit, supporting the presence of hydrodynamic outflows. None have yet precisely identified the physical extent of the out-of-transit signal, either due to non-continuous or short-duration observations. This strongly limits our ability to measure accurate mass-loss rates and to understand how the stellar environment shapes outflows. Here we present the continuous, full-orbit helium phase-curve observation of an exoplanet: the ultra-hot Jupiter WASP-121 b, obtained with the James Webb Space Telescope (JWST) and the Near Infrared Imager and Slitless Spectrograph (NIRISS). We detect significant helium absorption at $>3\sigma$ over nearly 60% of the orbit, revealing a persistent and large-scale outflow. The signal separates into a dense leading tail moving toward the star and a trailing tail pushed away by stellar irradiation. Both appear to remain collisional far from the planet, implying strong hydrodynamic escape. While qualitatively consistent with theoretical expectations, current models cannot reproduce the full spatial and kinematic structure, limiting precise mass-loss estimates. These results demonstrate JWST's ability to map exoplanet outflows in detail and highlight its synergy with ground-based spectroscopy.

We observed from October 26 to October 28 of 2023, the full red-optical to near-infrared phase curve of the canonical ultra-hot Jupiter WASP-121 b with the JWST using the NIRISS instrument[1] in the SOSS mode[2]. With a mass of 1.18 $M_{Jup}$ and an inflated radius of 1.87 $R_{Jup}$, WASP-121 b revolves around its F6-type host star every 1.275 days. Its proximity to the host star yields an equilibrium temperature of 2350 K, assuming zero Bond albedo and full heat redistribution[3]. Among the key results obtained for WASP-121 b, atmospheric escape has recently been detected through the metastable helium triplet[4,5] with CRIRES+[6]. This detection was obtained during transit, along with a slight hint of

**Fig. 1 | JWST/NIRISS excess absorption time series around the helium triplet relative to mid-transit time in the stellar rest frame.** The red dashed lines represent the transit ingress and egress, while the two secondary eclipses ingress and egress are shown with the orange dashed dotted lines. Clear excess is detected as the dark blue region not only during the full transit, but also before and after, for a large fraction of the orbit, suggesting an extended outflow. The color scale was truncated to −500 to +500 ppm for better visualization purposes.

pre-transit absorption. However, no precise constraints were set on the extent of the physical outflow. Throughout the paper, we used the system parameters from ref. 7. The phase curve was observed as part of the NEAT Guaranteed Time Observation program 1201 (P.I.: D. Lafrenière) and contains more than a complete orbit of the planet, capturing two secondary eclipses and a primary transit, for a total of 36.9 hours and 3452 integrations. We used the SUBSTRIP256 subarray, which captures the first and second diffraction orders of NIRISS/SOSS ($\lambda = 0.6$–$2.85\,\mu$m), and acquired 6 groups per integration to ensure all pixels remain below 80% full well capacity.

The NIRISS/SOSS observations were reduced using the `NAMELESS` data reduction pipeline following the methodology applied to past JWST observations (e.g., refs. 8,9). We apply special care to the treatment of the $1/f$ noise ($f$ is frequency) for this analysis of the metastable helium signal, as improper correction may lead to a dilution of the signal (see Methods and ref. 8). An independent data reduction is performed using the `exoTEDRF`[10–13] pipeline to ensure our results are robust to data reduction assumptions.

We perform an in-depth analysis of the helium signature throughout the full phase curve of WASP-121 b following the methodology described in refs. 14,15. Specifically, we study the relative flux of a pixel of interest, centered on the helium triplet (10836.34 Å, hereafter called the central pixel), by comparing it to the flux observed in its neighboring pixels (62 pixels surrounding the helium triplet have been selected to ensure a good baseline, from 10547.04 to 11127.17 Å). The analysis is done on the phase curves at the pixel resolution of JWST/NIRISS ($R$ ~600 at 1.1 μm). The first step consists of normalizing each phase curve by its average flux during the secondary eclipse. The choice of the average flux during the secondary eclipse is made to minimize contamination from the helium signature leading and trailing the planet, as we assume no a priori knowledge on the extent of the helium signature, as shown in refs. 16,17 with exospheres or outflows potentially extending far from the planet. We proceed to build a reference phase curve by averaging pixel phase curves outside the helium triplet, from 10547.04 to 10714.83 Å and from 10958.12 to 11127.17 Å (see Supplementary Fig. 1). The following step consists in dividing out each individual pixel phase curve by the aforementioned reference phase curve to remove the thermal emission and average transit signal surrounding the helium line, which are quasi-constant over this short wavelength range, and express the flux variation in excess absorption. The time series is then binned into 60 temporal bins of 37 minutes each.

Figure 1 presents the excess absorption time series around the helium triplet. A clear excess absorption is detected during transit in the central pixel containing the helium triplet position. In addition, absorption is detected both before and after transit over a significant fraction of the time series. To better estimate the properties of the helium signature, we co-add the time series into the time domain to study the pixel phase curves (Fig. 2) and into the spectral domain to study the planetary spectrum integrated over time (Fig. 3).

## Results

We identify three distinct escape dynamical regimes from the excess absorption phase curves of the helium triplet (pixel at 10836.34 Å), as shown in Fig. 2. As suggested by Fig. 1, helium absorption is well detected during transit with an average amplitude of 2778 ± 82 ppm (34$\sigma$) between $t_2$ and $t_3$ (Table 1). In addition to the absorption during transit, we detect absorption leading and trailing the planet (see Supplementary Fig. 3 for the detection level of the 60 temporal bins). Pre-transit absorption is detected in individual temporal bins over 3 $\sigma$ for up to 4.3 hours before ingress. The absorption increases quickly in less than one hour before plateauing at 934 ± 70 ppm for ~2 hours. Post-transit absorption is detected at more than 3 $\sigma$ for up to 9.3 h after egress. The absorption is steady around 701 ± 42 ppm for seven hours before slowly decreasing. We thus define three regimes: leading from −5.8 h to −2.7 h from mid-transit[7], transit from $t_2$ (−1.1 h) to $t_3$ (+1.1 h), and trailing from +2 h to +10.8 h from mid-transit. These three regimes are separated by clear transitions in the helium phase curve, corresponding approximately to the altitude where the planet's thermosphere extends beyond the Roche lobe and turns into tails. Due to the low spectral resolution of JWST NIRISS/SOSS and its PSF spreading across multiple pixels, the helium triplet is diluted and measurable in the two closest spectral pixels at bluer (10826.98 Å) and redder wavelengths (10845.70 Å)—hereafter called blue and red pixels. Furthermore, in the central pixel, the helium triplet is offset by −3 Å (−80 km s$^{-1}$) with respect to the median wavelength of the pixel (10836.34 Å). As a consequence, and assuming the helium signature is a Gaussian profile with no velocity offset to its rest position (10833.31 Å), more absorption is expected in the blue pixel than in the red pixel. In the leading regime, the blue and red spectral pixels show similar amplitudes and temporal behavior—evidence for the helium signature being red-shifted, tracing material that infalls at high velocities onto the star. In contrast, in the transit and trailing regimes, the blue pixel absorbs more than the red pixel. Furthermore, during the

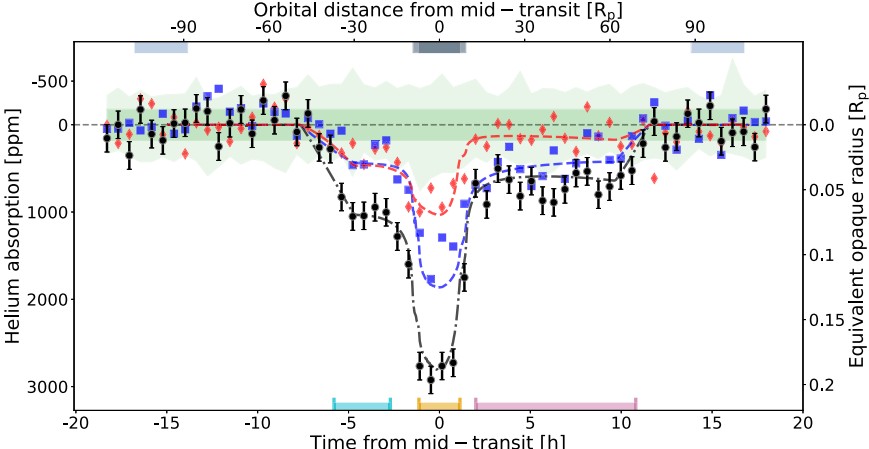

**Fig. 2 | JWST/NIRISS excess absorption phase curve around the helium triplet.** Binned into 60 temporal bins showing the helium absorption in part per millions (and its equivalent opaque radius in planet radius on the right axis) as function of the time from mid-transit in hours (and its equivalent distance in planetary radius on the top axis). The central pixel (10836.34 Å) is shown as the black dashed data points (with $1\sigma$ error bars). The two neighboring pixels of helium are shown in blue squares (10826.98 Å) and red diamonds (10845.70 Å). The dashed lines denote the best-fit toy model. The dark green region is the standard deviation of all the remaining pixels studied (from 10547.04–10714.83 Å and from 10958.12 to 11127.17 Å). In contrast, the light green region represents their upper and lower envelopes (obtained as the minimum and maximum flux value at each temporal bin). At the top, the two eclipses are indicated by the light blue-gray regions, while the transit is shown by the gray regions with ingress and egress as the light gray regions. At the bottom, the temporal signature is decomposed as the leading (cyan), transit (orange), and trailing (pink) signatures, same colors as Fig. 3. Source data are provided as a Source Data file.

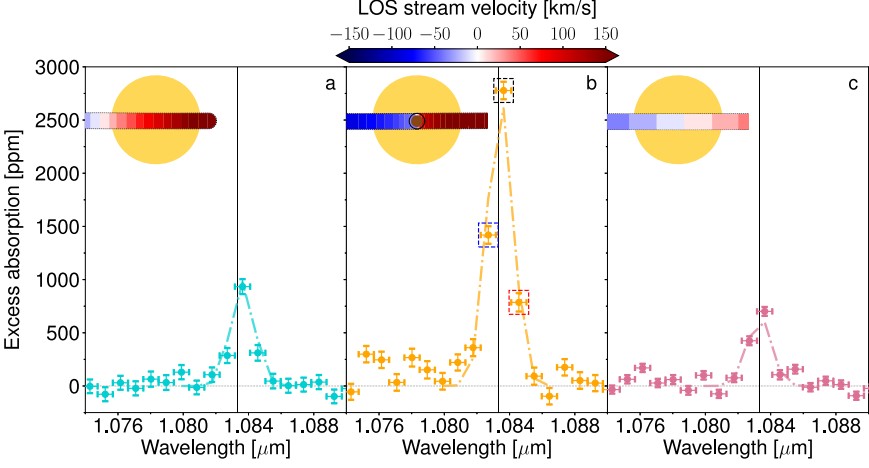

**Fig. 3 | JWST/NIRISS transmission spectra (with $1\sigma$ error bars) of the three regimes observed.** The leading, transit, and trailing spectra are shown in panels **a**, **b**, and **c**, respectively. The helium triplet wavelength position is shown as the gray vertical line. The dashed lines denote the best-fit toy model. In the three panels, we show the streams passing in front of the star at the middle of the corresponding phases. The velocity scale includes the planetary orbital motion and the outflow dynamics. The leading spectrum is clearly red-shifted, whereas the transit and trailing phases have contributions from both blue- and red-shifted stream elements. In panel **b**, the central pixel and its two neighboring pixels are framed by dashed squares, with colors matching those in Fig. 2. Source data are provided as a Source Data file.

**Table 1 | Measured absorption (with $1\sigma$ error bars) of the helium signature as a function of the three main regimes in the central pixel and its two neighboring pixels**

|  | Leading | Transit | Trailing |
|---|---|---|---|
| Blue pixel–10826.98 Å [ppm] | 286 ± 70 | 1419 ± 82 | 425 ± 42 |
| Central pixel–10836.34 Å [ppm] | 934 ± 70 | 2778 ± 82 | 701 ± 42 |
| Red pixel–10845.70 Å [ppm] | 311 ± 73 | 786 ± 86 | 105 ± 44 |
| Ratio blue/central | 0.31 ± 0.08 | 0.51 ± 0.03 | 0.61 ± 0.07 |
| Ratio red/central | 0.33 ± 0.08 | 0.28 ± 0.03 | 0.15 ± 0.06 |
| Ratio red/blue | 1.09 ± 0.37 | 0.55 ± 0.07 | 0.25 ± 0.11 |

Ratios between pixels (with $1\sigma$ error bars) are also provided.

transit regime, the central pixel absorbs twice as much as the blue pixel, while in the trailing regime, the absorption of both pixels is comparable–evidence of velocity offset. We further observe evidence for dynamics within the outflow structure. To quantify the variation in shape and amplitude of the helium signal over time, we produce transmission spectra for the three regimes by averaging their absorption, as shown in Fig. 3. Table 1 reports the absolute and relative absorption for the blue, red, and central pixels over the three temporal regimes. The leading regime shows absorption in the central pixel that is ~230 ppm deeper compared to the trailing regime, meaning that the number density of metastable helium atoms ahead of the planet is higher than those that are trailing. We also confirm the change in the line profile morphology as a function of time with increasing absorption in the blue pixel relative to the red pixel. Altogether, this

corresponds to the outflow being spread on both sides of the planetary orbit, with a component moving toward the star and a component blown away and escaping in a trailing tail shape (e.g., refs. 18–20).

## Discussion

Our ~17-hour continuous absorption of helium spans ~54% of the orbital revolution of WASP-121 b, making it the longest continuous orbital coverage of absorption from an extended atmosphere, only accessible due to the uninterrupted time series observations with high sensitivity at the helium triplet wavelengths enabled by the JWST. The physical extent of the material is equivalent to 107 planetary radii or 0.1 astronomical units along the planetary orbital motion. We distinguish three different regimes that can be attributed to distinct outflow structures. The leading regime is composed of a rapidly increasing helium density outflow with evidence of material being accreted on the star and starting a bit before the L4 Lagrange point. The transit regime is dominated by the extended thermosphere overflowing the Roche Lobe radius (1.3 $R_p$) and connected to the leading and trailing outflows. The trailing regime lasts for approximately a third of the revolution of WASP-121 b and is composed of a steady outflow that slowly decreases in density away from the planet and gets increasingly blue-shifted, likely due to pressure from the stellar wind.

Following the analyses made in refs. 14,15,21–23, we estimate the amplitude of the signature that would be observed at high resolution (~100,000) by considering a Gaussian model of fixed width (1 Å, in line with the CRIRES+ measurement[6]) centered on the helium triplet line position (10833.22 Å), convolved at the NIRISS/SOSS native resolution and fitted to the in-transit transmission spectrum. We retrieve an expected amplitude of 4.5 ± 0.6% (equivalent to ~2.0 $R_p$ in an optically thick regime), easily within reach of current facilities over one transit as demonstrated with NIRPS[20,24] simulated data in Supplementary Fig. 5. As we don't know the true width of the resolved helium profile, a correlation exists between the width, the amplitude, and the instrumental convolution. Supplementary Fig. 6 shows how the expected amplitude varies as a function of the assumed width. Furthermore, to properly retrieve the depth and line shape of the helium triplet at high resolution, one must use a reference spectrum taken around the secondary eclipse, where the effect of leading and trailing absorption is minimal. This is not the case in most observational setups, with short pre- and post-transit baselines generally observed. We further discuss and generalize the bias induced by the leading and trailing signatures in the characterization of the extended atmosphere in the methods.

While observations of the metastable helium triplet are generally performed in transmission (e.g., refs. 19,20,25–32), emission from the extended atmosphere is possibly detectable near secondary eclipse (e.g., refs. 33,34), when the star occults a significant portion of the extended outflow. Following the approximation for the amplitude of the helium airglow derived in ref. 33, our measured excess absorption of 2778 ± 82 ppm in-transit corresponds to a planet-to-star flux ratio of 47.8 ± 1.4 ppm. This is below the level of precision of what can be achieved at the pixel resolution (see Fig. 2). Observations at high spectral resolution, where the contrast is more advantageous (expected amplitude of 1350 ± 165 ppm), could provide additional constraints on the outflow structure and temperature. However, reaching this precision on a single line requires either 50 time-series events on a 4-m class telescope or a single event on a 39-m class telescope, like the ELT.

As one of the most studied exoplanets, WASP-121 b has benefited from numerous observations. The planet fills 59% of its Roche lobe volume, is aspherical due to tidal forces[3], and orbits its relatively young[35] (~1.1 Gyr) host star on a polar orbit[7]. Its atmosphere has been under intense scrutiny with the detections of metals[36–40] in its lower layers, alkali species[41,42] in its thermosphere, and ions[43], hydrogen[41], and helium[6] in its upper atmosphere. The metastable helium detection reported by ref. 6 was obtained from CRIRES+ transit observations and revealed an excess absorption of ~2%. This detection was obtained

during transit, along with a slight hint of pre-transit absorption. The discrepancy with our results in the signature depth and duration is a consequence of the baseline-induced bias described in the methods. Recently, a new study[44] made with the largest optical telescope, the four telescope units of the VLT, combined with ESPRESSO[44] resolved the unique vertical structure of the atmospheric dynamics of WASP-121 b. The lower atmosphere, probed by iron lines, is subject to day-to-night side winds, the stratosphere, probed by sodium lines, has an equatorial jet stream, and the thermosphere, probed by Hα, is prone to radial winds. In Supplementary Fig. 8, we confirm the detection (in agreement with refs. 41,44) of the Hα line with NIRISS/SOSS during transit, but we do not detect extended outflows for this hydrogen transition. We fitted a Gaussian to the transmission spectrum around Hα with an amplitude of 1265 ± 293 ppm (4.3σ), two times lower amplitude than the helium line, but with an uncertainty four times greater than the uncertainty of the helium line. Furthermore, the derived uncertainty from the Gaussian is slightly larger than the average pixel uncertainty (208 ppm) and than the standard deviation of the transmission spectrum continuum (227 ppm). The Hα light curve does not reveal significant absorption outside of the transit, and we set a 1-σ upper limit of 351 ppm over temporal bins of 74 minutes on the presence of extended outflow. While it is unlikely for hydrogen not to escape along helium particles, the non-detection of an outflow can be explained by the significantly lower throughput of NIRISS/SOSS order 2 at 0.65 μm (leading to larger scatter), and because it is possible that escaping hydrogen particles are not populating the Hα state. WASP-121 b is one of the first exoplanets for which we can study the complex dynamics of an exoplanet's atmosphere from the depths of its atmosphere up to its escaping materials. The strong outflow probed by our helium measurement likely drags heavier material, such as ions and alkali metals, into space.

The main avenues to properly model such extended helium outflow remain limited. On one side, there are 3D particle models simulating exospheres and cometary-like tails such as EvE (e.g., refs. 16,45). These models have shown their capability to reproduce helium signatures probing the exosphere (e.g., refs. 19,20,28,30,46) but are not able to self-consistently model extended tails in a fluid regime. Indeed, they assume a collisionless regime for the escaping gas, limiting the outflow extension to the short lifetime of these particles (about 2 hours for natural decay). On the other hand, 3D hydrodynamical codes can model such dense, extended helium outflows in a fluid regime from exoplanets, but appear to miss some key components to fully reproduce the observed amplitude. WASP-121 b orbits very close to its host star and, thus, is strongly subject to its gravity. As such, ref. 47 argued that Roche lobe overflow is necessary to explain the broadening of the observed escaping metal lines by ref. 43, suggesting a large mass-loss rate. Moreover, ref. 48 has predicted that the intense tidal forces on this planet would shape the outflow to streamlines along the planetary motion. We constructed a toy model in the EvE framework to account for a hydrodynamical regime forming elongated tails, approximated by two tubes. We fitted independently the morphology of the leading and trailing tubes that reproduce a tail-like structure. We provide in Figs. 3 and 4 schematics of the best-fit toy model to illustrate the spatial extension of the helium outflow and its dynamics as revealed by the JWST, which describes to first order the structure and dynamics of the planetary outflow. Especially, the similar level in the blue and red pixels in the leading regime constrains the velocity of the leading tail to ~90 km s⁻¹ towards the star. Interestingly, the first-order approach we used qualitatively reproduces the shape of the outflow presented in ref. 6, although their spatial extent appears smaller due to a limited baseline. In addition to tidal interactions, future models attempting to simulate in a self-consistent manner the leading and trailing tails of WASP-121 b and derive accurate mass-loss rates should account for asymmetries of the extended thermosphere at the base of the outflow, variations in sound speed and temperature within the outflow, and the

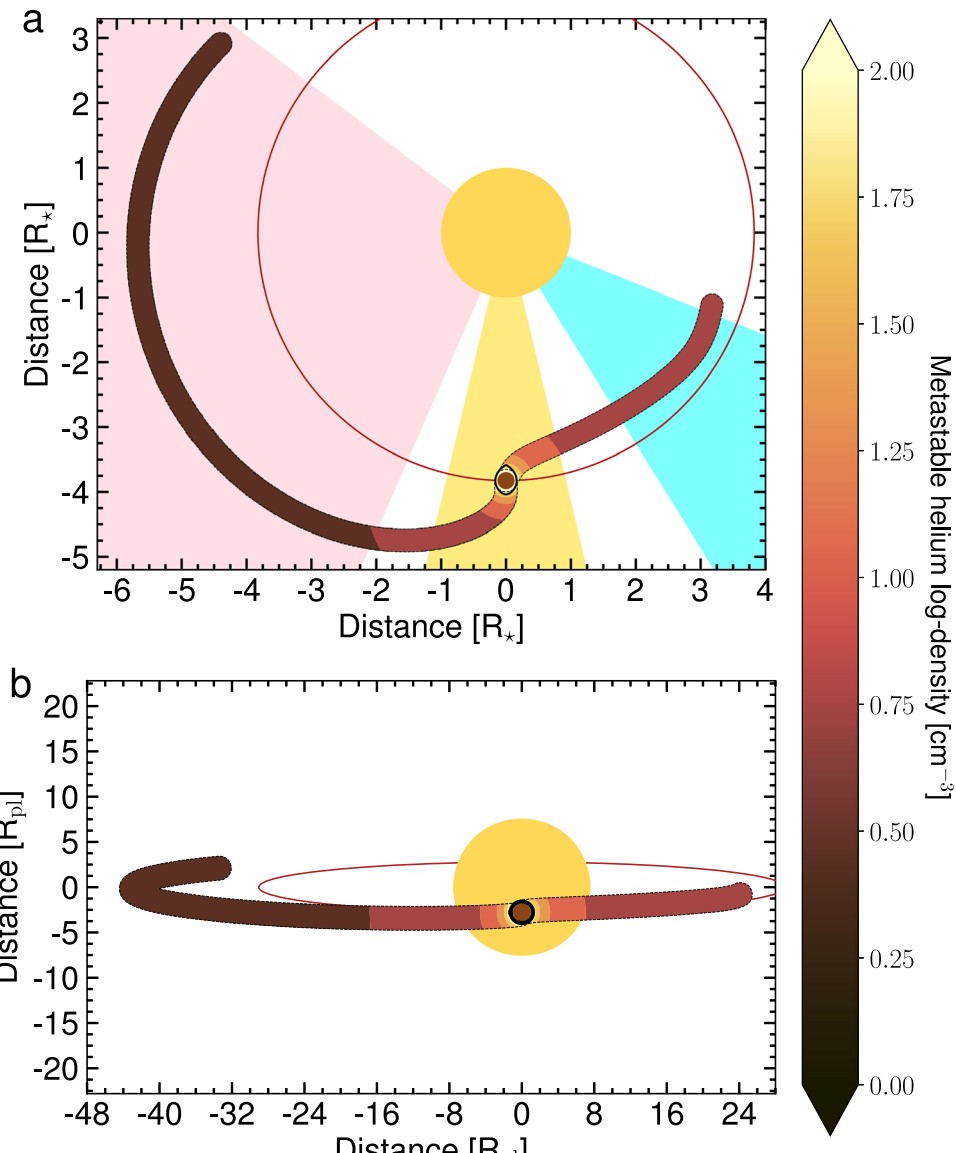

**Fig. 4 | 3D view of the best-fit toy model of the helium signature.** Above (**a**) and line-of-sight (**b**) views of the system. The solid red line indicates the orbital track of the planet, the black circle of the Roche lobe, and the brown disk of the planetary body. The three highlighted phases correspond to leading, transit, and trailing regimes, with similar colors to Fig. 2.

pressure imparted by the stellar wind and radiation pressure, all of which may significantly impact the shape and dynamics of the tails (see refs. 17,49). Nonetheless, given the red-shifted pre-transit signature, it appears that WASP-121 b's outflow is dominated by tidal interactions with hints for day-to-night asymmetries, which might indicate a relatively cold outflow (ref. 50). This makes WASP-121 b the perfect test case to understand to what extent the different aspects of star-planet interactions contribute to atmospheric outflows. Our results highlight the need for further model comparison and the inclusion of more precise physics and chemistry to properly handle the high-quality dataset we unveil to the community.

WASP-121 b is joining a category of planets (along with GJ 436 b[16,18,51], GJ 3470 b[52], HAT-P-32 b[50,53] and HAT-P-67 b[32,50]) that have detections of outstanding long exosphere or outflow structures (see Table 2). The two warm Neptune-mass planets orbiting M dwarfs, GJ 436 b and GJ 3470 b, have been reported to have extended hydrogen escape and a significant mass-loss rate ($10^{8-12}\,\mathrm{g\,s^{-1}}$). GJ 436 b is the canonical example of the cometary-like tail structure, whereas GJ 3470

b is losing more mass given its stronger stellar irradiation and faster photoionization, limiting the extent of the tail. HAT-P-32 b and HAT-P-67 b orbit F-type stars, and helium outflows have been measured in their stellar rest frame from the ground. For both planets, data were collected on multiple nights, leading to a partial collage phase curve. The outflow structure of HAT-P-32 b lasts for ~12 h (23% of the orbit), with longer leading than trailing absorption. For HAT-P-67 b, the outflow structure is only measured from pre-transit to egress, with no evidence of trailing absorption. In contrast to these two planets, WASP-121 b has a significantly longer trailing regime that could result from its short orbital period and thus stronger stellar tides on the atmosphere. This is supported by the difference in the leading tail velocity shift between HAT-P-32 b and HAT-P-67 b and modeled in ref. 50. The pre-transit absorption of these two planets is blue-shifted, whereas WASP-121 b's phase curve shows a red-shifted leading stream, indicating an outflow moving away from the observer and towards the star. Unlike HAT-P-32 b and HAT-P-67 b, the extended outflow shape of WASP-121 b is strongly driven by stellar tides rather than a combination of stellar

**Table 2 | Tail extent measurements for different exoplanets**

| Planet | duration [h] | Fraction of orbital period | Extension in AU | Extension in $R_p$ |
|---|---|---|---|---|
| WASP-121 b | 17 | 0.54 | 0.09 | 107 |
| HAT-P-67 b | 36 | 0.32 | 0.13 | 130 |
| HAT-P-32 b | 12 | 0.23 | 0.05 | 53 |
| GJ 436 b | 13–28 | 0.20–0.44 | 0.04–0.08 | 211–454 |
| GJ 3470 b | 5 | 0.06 | 0.01 | 71 |

The values are derived for WASP-121 b from this work, for HAT-P-67 b from ref. 32, for HAT-P-32 b from ref. 53, for GJ 436 b from ref. 51, and for GJ 3470 b from ref. 52.

winds and tides. Consequently, the outflow of WASP-121 b covers over half of its orbital track, significantly longer than any other planet's outflow. Other planets (e.g., WASP-69 b[20,26], WASP-52 b[15,54] and WASP-107 b[28,55]) might be susceptible to join this category of planets as their current helium observations reveal evidence for pre- and/or post-transit absorption, but a lack of temporal coverage limits our understanding of the complete extent of their atmospheric escape.

Our results call for a complete review of helium detections to properly assess the presence of pre- and post-transit absorption, which might be more frequent than expected and has been generally overlooked. This is particularly critical for ground-based high-resolution observations, as the current strategy commonly relies on short baseline measurements. Therefore, providing complementary observations to transits at different phases would assess the extent of exospheres and outflows. Such a technique has been used to measure the helium outflows of HAT-P-67 b[32] and HAT-P-32 b[53]. Moreover, this work outlines the unprecedented performance of JWST in probing extended helium signatures at large orbital distances from the planet's transit due to its high sensitivity and stability, leading to the measurement of strong dynamical velocity shifts. Helium phase curve observations covering a large sample of exoplanets are needed to fully understand the impact of escape on their atmospheric evolution and composition.

## Methods
### NAMELESS NIRISS/SOSS data reduction and spectroscopic extraction
We reduce the NIRISS/SOSS time series observations of the phase curve of WASP-121 b using the `NAMELESS` pipeline[8,9,11]. We start from the raw uncalibrated data and go through all Stage 1 and Stage 2 steps of the STScI `jwst` pipeline v1.12.5[56] to perform superbias subtraction, reference pixel correction, linearity correction, jump detection, ramp fitting, and flatfield correction. We skip the dark current subtraction step in Stage 1 as the latest reference file (jwst_niriss_dark_0171.fits) shows signs of leftover 1/f noise. We then proceed with a series of custom reduction steps to correct for bad pixels, non-uniform background, cosmic rays, and 1/f noise, following similar methodologies to those presented in ref. 9.

We correct for bad pixels by dividing the detector into a series of $4 \times 4$ pixel windows for each integration, in which each pixel whose mixed spatial second derivative ($\partial^2 f/\partial x \partial y$) is more than $4\sigma$ away from the median of its window is flagged as bad. Any pixel that is flagged in more than 50% of all integrations is then considered a bad pixel. These pixels are then interpolated over for all integrations using the bicubic interpolation function `scipy.interpolate.griddata`.

To correct for the non-uniform background, we follow the methodology described in ref. 57 and split the model background provided by STScI (https://jwst-docs.stsci.edu/) into two distinct sections separated by the jump in background flux that occurs around spectral pixel x~700. We then scale each section separately using portions ($x_1 \in [500, 650]$, $y_1 \in [230, 250]$) and ($x_2 \in [740, 850]$, $y_2 \in [230, 250]$) of the detector and using as the scaling factor the 16th percentile of the ratio of the median frame to the model background. These two portions were chosen to compute the scaling factors as they

show the least contamination from the spectral traces and background contaminants.

After subtracting the scaled model background from all integrations, we look for any remaining cosmic rays not flagged by the `jwst` pipeline jump detection step. To avoid flagging outliers caused by 1/f noise as cosmic rays, we first perform a temporary correction of the 1/f noise by subtracting the median frame scaled by the white-light curve (obtained by summing the counts from pixels $x \in [1200, 1800]$, $y \in [25, 55]$) from all integrations, and then subtracting the median from all columns. After this preliminary 1/f correction, we then compute the running median in time of all individual pixels and flag any pixels whose count is $3\sigma$ away from its running median at a given integration. These counts are subsequently clipped to the value of their running median. We finish this step by adding back the 1/f noise that was removed before the cosmic rays correction, and leave a more thorough correction of the 1/f to a later stage.

We measure the position of the center of the spectral orders at each column by cross-correlating the detector columns of the median frame with a template trace profile. For the template trace profile, we use column $x = 2038$ of the order 1 trace model contained in reference file `jwst_niriss_specprofile_0022.fits`. We then cross-correlate this template with all detector columns (using the `scipy.-signal.correlate` function), where we super-sample both the template and the data by a factor of 1000, and take the position of the maximum of the cross-correlation function as the center of the trace at a given column. Performing this for a first iteration returns the position of the trace of the first spectral order, we then repeat this procedure a second time, having masked region ($x \in [0, 1200]$, $y \in [130, 155]$) of the detector as well as all pixels within 30 pixels of the center of the trace, and obtain the position of the second spectral order trace from $x = 600$–1775.

We corrected for the 1/f noise using the same methodology presented in ref. 9. This method allows scaling independently all columns of each spectral trace, which is especially important for a phase curve and single line spectroscopy since the planetary flux is likely to show strong dependence on wavelength[8,9]. We correct for the noise by assuming that the counts $c_{i,j,k}$ of a pixel at integration $i$, column $j$, and row $k$ can be described as $c_{i,j,k} = \bar{c}_{j,k} s_i + n_i$, where $\bar{c}_{j,k}$ is the median column in time, $s_{i,j}$ is the scaling of column $j$ at a given integration $j$ and $n_{i,j}$ is the value of the 1/f noise for a given column and integration. We then minimize the $\chi^2$ of the fit to $c_{i,j,k}$ to obtain the best-fitting values of the scaling and 1/f. Finally, separate scaling values are applied to the spectral order 1 and 2 traces, considering only pixels that are within a 30-pixel distance from the center of the traces when computing the $\chi^2$, as 1/f shows important variation over the length of a single column.

After having performed all corrections on the detector images, we proceed to extract the flux from the first and second spectral orders using a simple box aperture. We extract the flux for box widths of 30, 36, and 40 pixels and find that they all result in similar (within ±2%) values of the point-to-point scatter for all wavelengths. We thus proceed with a box width of 40 pixels to minimize our sensitivity to variations in the trace morphology[8,9]. For the wavelength solution, we use the `PASTASOSS` Python package[58] with, as

input, the pupil wheel position that was recorded for these observations of WASP-121 b (245.766°).

Upon inspection of the reference white-light curve (Supplementary Fig. 1), we observe that the bottom of the two secondary eclipses does not occur at the same flux level. This is most likely caused by long-term stellar variability and instrument systematics in the data. However, the helium light curves have been corrected for these effects, as they have been divided by the reference white-light curve. Any chromatic variations in the stellar variability, instrument systematics, and planetary signal (except for the helium) should be negligible over the 0.06 μm-wide wavelength range that is considered in the analysis.

We also observe an apparent asymmetry in the shape of the primary transit. Possible scenarios for this asymmetry include uncorrected instrumental and astrophysical systematics that affect the transit shape on short timescales, or the presence of a significant temperature and molecular gradient between the morning and evening limbs of WASP-121 b (e.g., ref. 59). Another possibility is that the outflow of the planet, which we have found to be asymmetric, is optically thick to a certain extent and also impacts the transit shape outside of the metastable helium line. A robust verification of these hypotheses will require further modeling work.

### exoTEDRF

A detailed discussion of the data reduction and validation can be found in ref. 60. Here, we summarize the key aspects for completeness and performed a comparison with NAMELESS in Supplementary Fig. 2. We utilized exoTEDRF version 1.4.0 and closely followed the methodology outlined in refs. 11,12,61, alongside the official exoTEDRF tutorial (https://exotedrf.readthedocs.io/en/latest/content/notebooks/tutorial_niriss-soss.html). The reduction process began with Stage 1, where we opted to skip the dark current subtraction step to prevent the introduction of unnecessary noise.

To correct for $1/f$ noise, we applied the correction at the group level, prior to ramp fitting, by subtracting the background signal from each group. The background was removed following the same methodology as NAMELESS and scaling 2 regions about $x \sim 700$ from the STScI SOSS background model. Specifically, we employed the default exoTEDRF scaling for two background sections: ($x_1 \in [350,550]$, $y_1 \in [230,250]$) and ($x_2 \in [715,750]$, $y_2 \in [235,250]$). These regions were chosen to be sufficiently distant from contaminants and to effectively remove the background.

Once the background was subtracted, we performed the $1/f$ noise correction by constructing difference images. This was done by subtracting a median-stacked image from each frame and subsequently removing the median value from each column. To prevent bias in the $1/f$ correction, we carefully masked out contaminated and bad pixels. The initial mask, generated automatically by exoTEDRF, masked the spectral traces and identified bad pixels. Furthermore, we manually mask contaminants in the regions $x \in [0,400]$, $y \in [100,256]$ and $x \in [1050,1175]$, $y \in [100,256]$. Following $1/f$ correction, the background contribution was reintroduced to allow for flatfield and linearity corrections, after which the background was removed once more for the final time.

The centroids for each spectral trace were found using the edgetrigger algorithm[61]. We used a simple box aperture for spectral extraction and leveraged exoTEDRF's built-in optimization to determine the ideal pixel width. A width of 29 pixels was found to minimize scatter, but we ultimately selected 30 pixels for consistency in our analysis.

### Impact of stellar chromosphere contamination

We consider the extreme case, where the transit of WASP-121 b only occults quiet stellar regions with no production of metastable helium. This would result in an increased transit depth and the creation of a stellar pseudosignal. To assess the impact of stellar contamination on our measured signature, we followed the methods described in ref. 25. We assumed a helium equivalent width for the stellar line of 0.096 Å (estimated from CRIRES+ observations;[6]) and the NIRISS pixel size of 9.35 Å. We derive an upper limit on the stellar contamination of ~154 ppm, which is ~6% of the measured helium signature during transit. Furthermore, during the leading and trailing regimes, the opaque planetary disk is not in the line of sight and cannot induce a stellar pseudosignal. Therefore, only an extended transiting exosphere or outflow could induce a stellar pseudosignal, thus proving its existence.

An inhomogeneous stellar surface could also mimic a helium signature. The star WASP-121 has a rotation period of 1.13 days[3,7], similar to the planetary orbital period. Therefore, inhomogeneities at the stellar surface could create a helium signature. However, the amplitude of such a signature should remain constant over 37 hours (total phase curve duration) and therefore could not explain the helium amplitude variation that we are measuring. Furthermore, the equatorial velocity of WASP-121 is ~13.6 km s$^{-1}$[3,7], which is much lower than the velocity gradient derived with our model of the helium signature. Therefore, the measured helium signature is of planetary origin.

### Impact of center-to-limb variation and Rossiter-McLaughlin effect

The center-to-limb variation and Rossiter-McLaughlin effect are known to impact the line shape of spectral transitions present both in the star and the planet along the transit chord. Reference 62 studied their impact on the helium triplet on JWST/NIRSpec observations with the G140H and PRISM modes. Looking at the hot Jupiter HD 209458 b, they predict a bias of ~100 and 10 ppm for G140H and PRISM, respectively. The lower resolution of the PRISM causes the sharp decrease with respect to G140H. In our case, NIRISS/SOSS has a resolution of ~600. Therefore, despite WASP-121 b being a larger planet, we can estimate an upper limit on the helium signature of CLV and RM effects below 100 ppm, in the same order of magnitude as the 1−$\sigma$ uncertainties.

### Impact of extended pre- and post-transit absorption on retrieved properties

The 17-hour helium absorption reveals that the extraction of an extended signature has to be made with caution. Our first analysis of the transmission spectrum was obtained using a standard phase curve fitting routine using the batman package[63]. However, the phase curve parameters tried to adjust to the leading and trailing regimes to properly match the transit and secondary eclipse events. This led to a bias on the recovered transit depth of ~1200 ppm at the central pixel as shown in Supplementary Fig. 4. This difference induced by the phase curve model is a similar bias than assuming that there is no flux absorption just before ingress and after egress, as the plateau in the trailing regime might suggest, if the observations had been over a shorter temporal window. This calls for careful analysis of the outflow tracer with JWST and other instruments and, in particular, requires extended out-of-transit observations. Similarly, the same conclusion can be made for high-resolution studies. Due to observational constraints, most ground-based observations are limited to short pre- and post-transit observations of up to 1–2 hours. Moreover, ground-based studies are differential analyses due to the Earth's atmosphere's variability. Even if some studies have reported post-transit absorption signals, only a few have reported pre-transit absorption. This lack of detection comes mainly from the observational design of ground-based observations, centered around transits with limited baselines. For WASP-121 b this bias can reduce the amplitude of the resolved helium line by up to 2%, in agreement with the helium signature detected with CRIRES+[6]. Furthermore, the lack of a long out-of-transit baseline in the CRIRES+[6] explains why the outflow was not detected. With our result, we want to draw attention to the high interest in reanalyzing all high-resolution helium detections by extending the

baseline through new observations and, more importantly, to changes in the observational strategy of helium observations to estimate the occurrence of extended outflows.

## Stream modeling

We constructed a toy model to reproduce the observations, shaping the planet's outflow into leading and trailing streams. This assumption is motivated by the extreme orbital coverage of absorption as well as the surrounding stellar environment. Previous hydrodynamical simulations (e.g., ref. 48) suggest that Roche lobe overflows can form elongated structures along the planetary orbit when tidal interactions are strong. To approximate this structure, we constructed a tube following or preceding the planet in its orbit, with a radius perpendicular to the tube axis set to the Roche lobe radius. The tube-like shape in the planetary restframe is derived by solving the 2D gravitational equations for a free particle within the orbital plane, accounting for the planet and the star. The trajectory is parametrized by a launch velocity and direction relative to the planet's orbital motion. Given the asymmetric nature of Roche lobe overflows in tidally locked planets, we modeled the morphology of the 2 tails separately, fitting each with an outflow angle and a velocity at the Roche lobe. The resulting velocity field of the stream is then divided into two components: an orbital contribution from the planet's orbital velocity, and a gravitational component approximating the stream's proper motion within the orbital plane.

We used the EvE[16,45] code to perform simulations of WASP-121 b's transit, generating stellar flux time series at high spectral and temporal cadence, then convolving them to NIRISS/SOSS instrumental response. We then processed the synthetic flux series similarly to the data to extract the excess absorption signal. Firstly, we focused on the leading and trailing signatures. To reproduce this feature, the LOS surface density far from the planet has to remain nearly constant, albeit with different levels between the tails. We fitted the geometry of the two tails using the total absorption produced by the blue, red, and central pixels. It determined the spatial extent, and initial conditions of the outflow at the Roche lobe (velocity and direction of launch) in the orbital plane of each tail. We noticed at this point that it was impossible for our toy model to reproduce the ratio of blue and red pixels to the central pixel in the leading tail (see Supplementary Fig. 7). The motion towards the star of the leading tail was too strong compared to our toy model. We attribute this discrepancy to hydrodynamical effects such as pressure gradient, viscosity, or drag that may slow down the stream and orient it even more towards the star. For this reason, we also introduced a velocity gradient towards the star that scales as $1-1/r^2$ where $r$ is the distance to the planetary center in planetary radii. Finally, we modeled the core of the transit by injecting a hydrostatic density profile (the mean molecular weight being fixed to 1.3) within each tail, truncated at a given radius to avoid saturation near the planetary core. This profile produces a moderate and stable absorption far from the planet, which increases rapidly during the planet's transit.

Our best model successfully reproduces the initial and final phases of absorption in the observed phase curve (Fig. 2), with each pixel's intensity matching the data within the uncertainties. Notably, it estimates that the leading tail moves toward the star at ~90 km s⁻¹, while the trailing tail recedes at a few km s⁻¹. Additionally, our best-fit model suggests a higher outflow velocity and temperature at the Roche lobe for the leading tail than the trailing tail, which we attribute to the higher level of irradiation received by the dayside but must be investigated by self-consistent hydrodynamical simulations.

While our model accurately captures the transit phase for the central absorption pixel, it slightly overestimates the absorption levels in the blue pixel during transit. The discrepancy may be due to our approximated tail geometry, which reduces the extent to which the flow moves primarily together with the planet. As a result, the high-resolution absorption spectrum forms two distinct peaks during the

transit, spreading the signal into several JWST pixels due to the wide range of velocities the occulting elements of the stream have. In our framework, we set the stream radius equal to the Roche lobe, but in reality, the width of the outflow evolves significantly with the time of escape. We note, however, that high-resolution spectra produced by our toy model show similarities with ref. 50, in agreement with a planetary outflow temperature of a few thousand kelvin.

Finally, we also tried to model the outflow as a collisionless particle regime, or exosphere, similarly to ref. 64. However, we found no configuration capable of reproducing the observed plateau over such an extended duration. The reason is that XUV and natural decay of metastable helium produce tails with a fast-decreasing exponential density profile. As a result, the relatively rapid increase in excess absorption on one side, later followed by a smooth decrease, cannot be explained within a self-consistent collisionless framework without invoking large-scale hydrodynamical structures. This fully confirms to us the hydrodynamical nature of WASP-121 b outflow, which is strongly shaped by the tidal interactions with its host star. In this specific case, we cannot conclude about the mass loss of this planet with our toy model. A more rigorous treatment is required, also including stellar winds, or day-to-night side asymmetries, known to significantly influence the shape and dynamics of the outflow (see refs. 6,17,49,50). However, these quantities remain poorly constrained by current observations, limiting our ability to make precise predictions. Nonetheless, simplified models like the one presented here can provide valuable insights and help understand the overall structure needed to match observations for more advanced simulations.

## Data availability

This work is based on observations made with the NASA/ESA/CSA James Webb Space Telescope. The data were obtained from the Mikulski Archive for Space Telescopes (https://archive.stsci.edu/) at the Space Telescope Science Institute, which is operated by the Association of Universities for Research in Astronomy, Inc., under NASA contract NAS 5-03127 for JWST. These observations are associated with program #1201 and are publicly available through the following https://doi.org/10.17909/ppjp-9743) under GTO program 1201 (principal investigator D.L.). Source data are provided with this paper. All the data presented in the figures of this study are available at ref. 65 or through Zenodo: https://doi.org/10.5281/zenodo.16994852. Source data are provided with this paper.

## Code availability

This research made use of the Astropy[66–68], Matplotlib[69], NumPy[70] and SciPy[71] Python packages. The open-source codes that were used throughout this work are listed below: `batman` (https://github.com/lkreidberg/batman); `ExoTEDRF` (https://exotedrf.readthedocs.io/en/latest/); The reduction code `NAMELESS` (contact: Louis-Philippe Coulombe; louis-philippe.coulombe@umontreal.ca) and the modeling code EvE (contact: Vincent Bourrier; vincent.bourrier@unige.ch) are not publicly available, but can be accessed on request. The python scripts used to produce the figures are available at ref. 65 or through Zenodo: https://doi.org/10.5281/zenodo.16994852.

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

## Acknowledgements

This project was undertaken with the financial support of the Canadian Space Agency. R.A. acknowledges the Swiss National Science Foundation (SNSF) support under the Post-Doc Mobility grant P500PT_222212 and the support of the Institut Trottier de Recherche sur les Exoplanètes (IREx). This project has received funding from the European Research Council (ERC) under the European Union's Horizon 2020 research and innovation program (project SPICE DUNE, Grant Agreement No. 947634). This work has been carried out in the frame of the National Center for Competence in Research PlanetS, supported by the Swiss National Science Foundation (SNSF) under grants 51NF40_182901 and 51NF40_205606. J.D.T. acknowledges funding support by the TESS Guest Investigator Program G06165. D.J. is supported by NRC Canada and by an NSERC Discovery Grant. S.P. acknowledges support from the Swiss National Science Foundation under grant 51NF40_205606 within the framework of the National Center of Competence in Research PlanetS. C.P.-G acknowledges support from the E. Margaret Burbidge Prize Postdoctoral Fellowship from the Brinson Foundation. M.R. acknowledges support from NSERC and the Postdoctoral Fellowship program. L.D. is a Banting and Trottier Postdoctoral Fellow and acknowledges support from the Natural Sciences and Engineering Research Council (NSERC) and the Trottier Family Foundation.

## Author contributions

R.A. led the writing of this manuscript. R.A. led the data analysis and interpretation of the helium signature. L.P.C. led the data reduction with NAMELESS and wrote the related sections. Y.C. imagined and built the tail model implemented in the EvE framework developed by V.B. and performed the simulations to reproduce WASP-121 b He observations. Both Y.C. and V.B. contributed to the development of the tail model and the interpretation of the He signature reported in the paper. J.S. performed the data reduction and spectral extraction with Exo-TEDRF. D.L. is the principal investigator of the NEAT GTO program. L.A., E.A., D.L., and R.D. have contributed to the development of the instrument. R.D. is the principal investigator of the NIRISS instrument on board JWST. M. R. M. made significant contributions to the development of the instrument and played a key role in establishing the guaranteed time observing program. R.A., L.P.C., Y.C., J.S., L.D., V.B., D.L., L.A., E.A.B.B., N.B.C., R.D., V.K., R.J., D.J., A.B.L., M.R.M., S.P., C.P.G., M.R., J.T., and J.D.T. contributed to the interpretation and discussion of the results and provided significant comments and suggestions to the manuscript.

## Competing interests

The authors declare no competing interests.

## Additional information

[1]Institut Trottier de recherche sur les exoplanètes, Département de Physique, Université de Montréal, Montréal, QC, Canada. [2]Observatoire astronomique de l'Université de Genève, Versoix, Switzerland. [3]Trottier Space Institute at McGill, 3550 rue University, Montréal, QC, Canada. [4]Department of Earth and Planetary Sciences, McGill University, Montréal, QC, Canada. [5]Department of Physics and Astronomy, University of Waterloo, Waterloo, ON, Canada. [6]Department of Earth, Planetary, and Space Sciences, University of California, Los Angeles, CA, USA. [7]Department of Astronomy and Carl Sagan Institute, Cornell University, Ithaca, NY, USA. [8]NRC Herzberg Astronomy and Astrophysics, Victoria, BC, Canada. [9]Department of Physics and Astronomy, University of Victoria, Victoria, BC, Canada. [10]Department of Physics and Astronomy, Johns Hopkins University, Baltimore, MD, USA. [11]Department of Astronomy, University of Michigan, Ann Arbor, MI, USA. [12]Department of Astronomy & Astrophysics, University of Chicago, Chicago, IL, USA. [13]Department of Physics, University of Oxford, Oxford, UK. SNSF Postdoctoral Fellow. ✉e-mail: romain.allart@umontreal.ca

