## [Transparent Peer Review file · Nature Communications]

A complex structure of escaping helium spanning more than half the orbit of the ultra-hot Jupiter WASP-121b

Corresponding Author: Dr Romain Allart

Version 0:

Reviewer comments:

Reviewer #1

(Remarks to the Author)

I thought that this was a careful work of fundamental importance in the field of atmospheric escape from exoplanets. The primary finding is that excess absorption of metastable helium at 1083nm extends well beyond the planet along the path of the orbit. The authors convincingly argue that this morphology implies a hydrodynamic outflow shaped by tidal forces in the star-planet system. This is the first time such an outflow has been continuously monitored from a space-based observatory, offering a stable baseline for the observations. This work offers a new insight into how atmospheres "boil off" of exoplanets, and shows that current strategies for ground-based observations may need to be revised. I enthusiastically recommend that this paper could proceed toward publication with minor revisions.

I agree on the central importance of continuous space-based monitoring for stability and control of stellar variability. On the other hand, I felt that the distinction between WASP-121 b, HAT-P-32b and HAT-P-67b, was perhaps overstated in some ways.

For example, the argument was made that the two HAT planets have blueshifted He pre-transit. My understanding of the papers analyzing these data is that the absorption is spread around approximately the stellar rest frame, with slight red shifts immediately pre-transit and possible blue shifts at the extreme reaches of the tails. This seems in line with those tails being tidally-sculpted just like WASP-121b, which is the argument made by the authors of those papers as cited in the paragraph beginning on line ~204. Could the authors confirm that their intent is correctly stated and that it matches the statements being made in the papers cited.

I believe it would be interesting to discuss the hydrodynamic modeling in Ref 29 in terms of the toy model adopted. The approach appears consistent with the stream-like toy model adopted to analyze the JWST light curve. And the conclusions share the outcome of tidally-shaped tails.

I found the description of the data reduction quite thorough and comprehensive.

One aspect of the toy model that was unclear to me was the exact velocity and density profiles within the streams. I understood from the stream-modeling section in the Methods that the density is the composite of an exponential near the planet and an extended distribution. And that the line-of sight velocity influences the position. For reproducibility, could the authors quantify these statements about the toy model somewhat more? Either via the defining equations or in a figure? Fig 4 might be easier to read in a logarithmic color scale, but of course I leave that to the authors discretion.

typo: "At the difference of these two planets,"

Code availability: could the authors release the tools to reproduce their toy model along with this work? The light curve reduction is supported by the open source packages mentioned, but not the analysis/interpretation.

Finally, I want to congratulate the authors on a stunning observation and their careful investigation of its significance.

Reviewer #2

(Remarks to the Author)

I would like to apologize. I was sick, then I had too much work, and therefore I can submit my report too late.

The paper contains important work, nice results. Its value that the signal detected here is large, and there is no doubt that there is a very large spectroscopic signal,

The paper is well written, the Figures are beautiful.

I recommend the paper to accept it for publication, and I have just a few minor comments to improve it.

Instead of (or next to) Reference 13 which is about the aspherical shape of the star, I strongly recommend to cite Hellard et al. (2020), ApJ 889 66 which characterized better the shape of the planet based on HST observations.

Data availability: I strongly urge the authors not ONLY to give the MAST Archive as a reference. Please, in addition, publish your auxiliary files as well as your final, reduced, cleaned light curves what the authors used in their final analysis. This helps the other investigators to compare their reduction to your, and/or it helps other. non-experts of the JWST-instruments to do their own analysis.

Page 17: Can you give references to the cited data in the table caption, please?

Extended Data Fig. 1.: The two observed occultation event seems to be have different depth - at least if I compare the blue points to the black dashed line. This is probably due to systematics in the instrument and/or stellar variability. Can the authors, please, characterize the height and depth of the occultation effects, judge the difference in terms of sigmas between the two occultation events and write the uncertainties caused by this baseline-variation?

Extended Data Fig. 1.: the transit shape shows clear asymmetry for me. Is this true? If yes, can it be due to shape distortion as discussed by Hellard et al. (2020), ApJ 889 66?

Extended Data Fig 7.: I am always afraid of having a statement based on only one point. And the confidence region in that Figure is defined by the error bars of one point of H-alpha. Can the authors comment the reliability of the fit visible in that Figure?

Reviewer #3

(Remarks to the Author)

I carefully read the paper "A complex structure of escaping helium spanning more than half the orbit of an ultra-hot Jupiter" by Allart and co-authors. Overall, my impression is that the manuscript presents important new observational results, which are definitely worth publishing. The data analysis is solid and the methods are clearly explained with few exceptions. The modeling remains on the weaker side but is sufficient for a paper with a strong observational focus. The structure of the paper is a bit unusual as the authors jump into the data reduction and instrumental part almost immediately before discussing the object and the science a bit in what would typically be considered an introduction. I think it is a pity that the authors do not spend a bit more time on discussing the simulations detection of Ha absorption and how it compares to the He I detection in their paper. Below, I make a few comments and raise a few questions, which I think should be addressed before publication.

Title: I think the title should mention the object WASP-121.

Abstract:

"However, none of these observations identify the physical extent of the out-of-transit signal, either due to non-continuous or short-duration observations"

 The authors should be more precise on what they mean. In HAT-P-32, for instance, I think that observations did show the extent of the atmospheric structure (Zhang et al. 2023).

"display very different dynamics"

 I assume the two tails are meant. If so, name the key difference.

"they must be combined with JWST phase curves"

 While I agree that phase curves (or rather full phase coverage) is important, it is not vital that JWST is involved. HAT-P-32b is a case in point.

l47: a complete orbit  more than a complete orbit

l59: relative flux [...], to its neighboring pixel  This is understandable but not well formulated. For example say "Specifically, we study the relative flux in [...] by comparing it to the flux observed in neighboring pixels." or something

similar.

Figure 1: What is the reference frame for these spectra (planetary rest frame, stellar rest frame)?

I65: have no a priori  assume no a priori knowledge on the extent of the helium feature in WASP-121; there are (a) previous observations of WASP-121b and (b) observations of other systems, which the authors cite and use as a priori knowledge to justify their procedure, which is of course reasonable.

I72: Why do the authors use 37 min binning? This seems long compared to the optical transit duration.

I76, I81: I am confused about the authors' method of co-adding. Looking at Fig. 1, the absorption in the central pixel should be a bit more than 400 ppm during the optical transit. In Fig. 2, it appears to be a lot higher (average amplitude of 2778 +/- 82 ppm). It is not clear to me how this difference comes about. Likewise, I am not entirely sure what is shown in Fig. 3. Is it the sum of the individual absorption spectra or the average spectrum during the respective periods? The authors should clarify this.

I90: The authors use the term "phase curve" to refer to the temporal variation of the excess absorption as far as I understand. This might be confused with what is shown in Extended Fig. 1. I suggest to introduce unique terminology.

I94, Table 1: blue and red pixel become left and right in Table 1.

Table 1: The ratios should be given with uncertainties.

I108: "meaning that the number of metastable helium atoms ahead of the planet is higher than those that are trailing"  The authors should be more precise. I think this means that the number of metastable He absorbers in front of the star is higher before the transit than after it, assuming that conditions are reasonably optically thin. As for the total number before and after, the duration of the respective periods has to be considered. I suggest the authors produce a plot showing the time evolution of the mean wavelength of the absorption component.

I115: longest **continuous** orbital coverage  Assuming that the structure is reasonably stable in time, the continuous coverage is not actually crucial to study it although it is certainly a plus.

I121: I do not understand the significance of the L2 point here. I think the leading tail would be launched through the L1 point if such a simplification applies at all and the outflow is not approximately radial throughout a larger volume.

I151 and following paragraph: I think that most of the information given here would better be placed at the beginning of the paper. An upper limit on the out-of-transit non-detection of H α would be valuable information.

I173 and following paragraph: It may be worth mentioning that the simulations shown by ref 29 do qualitatively show the structures reported here although they are of course based on different assumptions about the absorption.

I191: "[...] directly probing atmospheric accretion by the host."  Motion toward the star at some point is no direct probe of accretion in my opinion.

I229: I don't think these have generally been overlooked. It is, however, true that most ground-based campaigns have to rely on short baselines for technical reasons.

I334: "However, the amplitude of such a signature should be rather constant over its line-of-sight visibility and not vary along the phase curve as we see."  I do not understand this conclusion. I think the authors state correctly that rotational modulation can introduce a signal. The amplitude of this signal could in the extreme be similar to the stellar line EW and likely much lower. Did the authors intend to say that the time variation of a potential stellar signal would not be expected to be aligned with the planetary transit timing? If so, I agree, but the authors should clarify their statement.

I336: "Furthermore, the equatorial velocity of WASP-121 is about 13.6 km s⁻¹ 13;14, which is much lower than the velocity gradient we measure in the helium signature."  Have the authors quantified the velocity gradient? At a spectral resolution of 600 (500 km/s) this sounds like a challenging task. However, as I suggested earlier, the barycenter of absorption might tell us something about the velocity shifts and that would actually be interesting information.

I350: While the experiences made by the authors during their fitting attempts are interesting, I suggest to focus on the results in this section. As the authors correctly point out, short baselines and the differential measurement in ground-based campaigns can lead to the described effects.

I361: "Even if some studies have reported post-transit absorption signals, it is always assumed that pre-transit data do not contain absorption signatures due to a lack of evidence until now"  As to my knowledge, this is simply not true. People have been aware of the possibility but were often forced to make such assumptions owing to the lack of more complete phase coverage. Also pre-transit absorption was reported previously, e.g., in HAT-P-32 b (even with a "conventional" short-baseline transit campaign, see Czesla et al. 2022 and with more complete orbital sampling by Zhang et al. 2023).

I366: "mandatory": Perhaps rather 'potential value of a reanalysis'.

I417: I think 29 should be cited here as well as this was actually discussed there for the specific case of WASP-121.

Extended Fig. 2: I think this figure is never referenced in the text.

Version 1:

Reviewer comments:

Reviewer #1

(Remarks to the Author)

Thank you to the authors for their thoughtful responses and revisions. These modifications have satisfied my concerns and in my view the paper is ready for publication. Congratulations on the excellent work.

(Remarks on code availability)

A brief readme file might be helpful, otherwise the code appears self explanatory and is a helpful addition to the manuscript.

Reviewer #2

(Remarks to the Author)

Dear Authors,

thank you for your answers and changes in the manuscript. I have no objection against publication. Congratulations to you work!

Best regards

(Remarks on code availability)

Reviewer #3

(Remarks to the Author)

I thank the authors for taking my critique into account. My points were addressed properly. The paper has improved in clarity, and I am happy to recommend it for publication after minor revision. I would like to congratulate the authors on their fantastic result.

Two minor points came to my attention in the revised version:

Fig. 1, caption) I suggest the authors further clarify that the scale was truncated w.r.t. that used in the other figures (as opposed to 'set') to highlight the smaller structures.

I321) I am very skeptical that gravitational darkening can introduce a strong enough asymmetry because the planetary orbit is nearly polar (leading to a symmetric signal) and although the star notably rotates, it isn't a very fast rotator like WASP-33 for example.

(Remarks on code availability)

Version 2:

Reviewer comments:

Reviewer #3

(Remarks to the Author)

I thank the authors for taking my comments into account.

The paragraph in question is correct, as far as I can say. As the claim of an asymmetric transit remains unsubstantiated by means of a rigorous analysis in this paper (it is "apparent" in the words of the authors), I suggest to remove the second sentence (about tidal distortion and gravitational darkening) all together and leave the reader with the possible explanations (for a possible effect). Actually, at least to my eye, this looks like a neat planetary phase curve, maybe indicative of a hot spot offset from the substellar point as has been observed in many systems before. I agree though that further modeling is required.

(Remarks on code availability)

Referee report

We thank the three referees for their positive reports and kind words. Their feedback improved the overall quality of the paper. We have considered all their comments and answered them below in blue. Modifications to the manuscript have been highlighted in blue as well.

Firstly, we would like to inform the referees that we have modified our modeling approach as follows. We updated the stream model to incorporate a better prescription of its shape. In the old model, the trajectory in the planetary rest frame followed by the stream was approximated by the orbital track. This trajectory was then rotated to account for tidal forces and outflow ejection asymmetry. However, the trajectory was roughly approximated and not in line with more advanced models. One of the highlights of the presented observations is the tidal interaction shaping the outflow at large distances from the planet. In this context, we changed the prescription of the stream trajectory. We solve numerically the restricted 3-body problem in the planetary rest frame and orbital plane, initialized by an angle and velocity of launch with respect to the orbital track. This stream shape is more similar to typical results obtained by 3D hydrodynamical models.

Second, we address their concerns about data availability in a single response. All the data presented in the figures, with the code used to produce them, as well as the final extracted light curves, will be made publicly available on Zenodo once the paper is published. Referees can find the preview link here: <https://zenodo.org/records/16994852?preview=1&token=eyJhbGciOiJIUzUxMiJ9.eyJpZCI6IjcyODFIYmZhLWQ5YTQtNGE3Zi04YTUzLWQ2YWQ0NzI1MGVjOSIsImRhdGEiOnt9LjYyZW5kb20iOiIyZTU2ZjdiNjI3MTg5ZiY0MjhhNTdmNzMyMDQ0MTg4YyJ9.cCjhQmSS7bwOKd6Vps2nYzdvwPpWqVO4GDRW9Kvh7zH2ioOjAHSr8MUKOh3vaFdMZFmGgXd58ykawXfwN-JJg>

The raw data will be made available through MAST. Regarding the codes used in this work, we agree that they should be publicly made available for reproducibility. The code used to reduce the data NAMELESS is in the process of being made publicly available on GitHub by its owner. We also want to point out that exoTEDRF, the second reduction pipeline used, is publicly available on GitHub. Regarding our toy model using the EvE code, its owners are putting substantial effort into making the code user-friendly and releasing a stable version. They are preparing a paper that summarizes all the latest updates and functionalities of the EvE code added over the years (the latest description of the code contents was published in 2016). In this regard, considering that the present paper does not focus on the EvE code, we request not to release the code at this time. Otherwise, there is a risk that EvE is misused due to the lack of documentation and the sheer number of options available through the code, weakening the impact of its release and limiting its diffusion within the community.

Below are the remaining points of their comments, addressed point by point.

Reviewer #1 (Remarks to the Author):

I thought that this was a careful work of fundamental importance in the field of atmospheric escape from exoplanets. The primary finding is that excess absorption of metastable helium at 1083nm extends well beyond the planet along the path of the orbit. The authors convincingly argue that this morphology implies a hydrodynamic outflow shaped by tidal forces in the star--planet system. This is the first time such an outflow has been continuously monitored from a space-based observatory, offering a stable baseline for the observations. This work offers a new insight into how atmospheres "boil off" of exoplanets, and shows that current strategies for ground-based observations may need to be revised. I enthusiastically recommend that this paper could proceed toward publication with minor revisions.

I agree on the central importance of continuous space-based monitoring for stability and control of stellar variability. On the other hand, I felt that the distinction between WASP-121 b, HAT-P-32b and HAT-P-67b, was perhaps overstated in some ways.

For example, the argument was made that the two HAT planets have blueshifted He pre-transit. My understanding of the papers analyzing these data is that the absorption is spread around approximately the stellar rest frame, with slight red shifts immediately pre-transit and possible blue shifts at the extreme reaches of the tails. This seems in line with those tails being tidally-sculpted just like WASP-121b, which is the argument made by the authors of those papers as cited in the paragraph beginning on line ~204. Could the authors confirm that their intent is correctly stated and that it matches the statements being made in the papers cited. From Fig.1 of ref. 49, we see that the tails of the two HAT-P planets are slightly blueshifted. Despite coming from the same underlying process (tidal interaction), we think that WASP-121b tails show an interesting, strong proper motion, which is not in line with the planetary orbital track. Whereas the two HAT-P planets show weaker

intrinsic motion within the tails, and no sign of outflow falling towards the star. In this regard, taking also the continuous nature of our observation into account, we make a distinction with these two planets. We modify a sentence (line 234-236) to clarify our point.

I believe it would be interesting to discuss the hydrodynamic modeling in Ref 29 in terms of the toy model adopted. The approach appears consistent with the stream-like toy model adopted to analyze the JWST light curve. And the conclusions share the outcome of tidally-shaped tails.

We are now mentioning this in the modeling section.

One aspect of the toy model that was unclear to me was the exact velocity and density profiles within the streams. I understood from the stream-modeling section in the Methods that the density is the composite of an exponential near the planet and an extended distribution. And that the line-of sight velocity influences the position. For reproducibility, could the authors quantify these statements about the toy model somewhat more? Either via the defining equations or in a figure? Fig 4 might be easier to read in a logarithmic color scale, but of course I leave that to the authors discretion.

We slightly modified the model and the "Stream modeling" (line 396) section to increase clarity.

typo: "At the difference of these two planets,"

Done

Reviewer #2 (Remarks to the Author):

I would like to apologize. I was sick, then I had too much work, and therefore I can submit my report too late. The paper contains important work, nice results. Its value that the signal detected here is large, and there is no doubt that there is a very large spectroscopic signal. The paper is well written, the Figures are beautiful. I recommend the paper to accept it for publication, and I have just a few minor comments to improve it.

Instead of (or next to) Reference 13 which is about the aspherical shape of the star, I strongly recommend to cite Hellard et al. (2020), ApJ 889 66 which characterized better the shape of the planet based on HST observations.

We thank the referee for this suggestion

Page 17: Can you give references to the cited data in the table caption, please?

Done

Extended Data Fig. 1.: The two observed occultation event seems to be have different depth - at least if I compare the blue points to the black dashed line. This is probably due to systematics in the instrument and/or stellar variability. Can the authors, please, characterize the height and depth of the occultation effects, judge the difference in terms of sigmas between the two occultation events and write the uncertainties caused by this baseline-variation?

We thank the referee for pointing this out. There are indeed stellar variability/instrument systematics in the white-light curve. However, these effects have been corrected for in the helium light curves since we divide the three light curves of interest by that of the neighboring wavelengths. Any chromatic variations in the stellar variability, instrument systematics, and planetary signal should be negligible over the ~0.06-microns-wide region that is considered for the analysis. To make this clear, we added a paragraph in the main text (lines 313-319).

Extended Data Fig. 1.: the transit shape shows clear assymetry for me. Is this true? If yes, can it be due to shape distortion as discussed by Hellard et al. (2020), ApJ 889 66?

We thank the referee for pointing this out. In principle, tidal distortion of the planet should not lead to an asymmetric transit light curve (e.g., Saxena+2014). We think that the most likely explanation for this effect is that of gravity-darkening from the star, which is a fast rotator. We now mention this in the text (lines 320-322).

Extended Data Fig 7.: I am always afraid of having a statement based on only one point. And the confidence region in that Figure is defined by the error bars of one point of H-alpha. Can the authors comment the reliability of he fit visible in that Figure?

We understand the referee's concerns, and we have provided further details (lines 171-179) on the significance of the H \$\alpha\$ detection and how its uncertainty compares to the standard deviation of the transmission spectrum continuum. We are also confident that we detect the planetary H \$\alpha\$ line, rather than a systematic effect, as its amplitude agrees with previous results obtained at high spectral resolution.

Reviewer #3 (Remarks to the Author):

I carefully read the paper "A complex structure of escaping helium spanning more than half the orbit of an ultra-hot Jupiter" by Allart and co-authors. Overall, my impression is that the manuscript presents important new observational results, which are definitely worth publishing. The data analysis is solid and the methods are clearly explained with few exceptions. The modeling remains on the weaker side but is sufficient for a paper with a strong observational focus. The structure of the paper is a bit unusual as the authors jump into the data reduction and instrumental part almost immediately before discussing the object and the science a bit in what would typically be considered an introduction. I think it is a pity that the authors do not spend a bit more time on discussing the simulations detection of Ha absorption and how it compares to the He I detection in their paper. Below, I make a few comments and raise a few questions, which I think should be addressed before publication.

We based our structure on other papers in the field published in Nature journals. We nonetheless added a line on the context around atmospheric escape for WASP-121b to the first paragraph.

Title: I think the title should mention the object WASP-121.

Done.

Abstract:

"However, none of these observations identify the physical extent of the out-of-transit signal, either due to non-continuous or short-duration observations"  The authors should be more precise on what they mean. In HAT-P-32, for instance, I think that observations did show the extent of the atmospheric structure (Zhang et al. 2023).

We agree that for HAT-P-32b and HAT-P-67b, absorption signatures outside of transit have been detected by Zhang+2023 and Gully Santiago+2024. However, the estimated duration of these out-of-transit signatures, and thus their physical extent, was not precisely estimated due to the non-continuous nature of their observations and the collage of multiple visits that other variability sources might impact. Here, for WASP-121b, the continuous nature of the JWST phase curve allows us to uniquely and precisely measure the duration of the outflow observations. We revised the text to better convey our intended meaning (line 26).

"display very different dynamics"  I assume the two tails are meant. If so, name the key difference.

Done

"they must be combined with JWST phase curves"  While I agree that phase curves (or rather full phase coverage) is important, it is not vital that JWST is involved. HAT-P-32b is a case in point.

JWST offers the unique advantage of measuring continuous, full-phase curves at unprecedented precision. In the case of HAT-P-32b (or HAT-P-67b), the compilation of multiple visits results in gaps in the time series that can only be filled by observing the target over multiple nights, thus spanning multiple weeks or months. This inherently leads to potential variability in the instrument, on the star, or even within the outflow. Furthermore, the photometric precision achieved by JWST is unbeatable compared to ground-based instruments. Finally, these three planets demonstrate that numerous aspects of atmospheric escape and outflow, measurable with the helium triplet, remain unexplained. Thus, JWST offers a more straightforward, more precise way to constrain the temporal properties of the helium signature. Still, it must be combined with high-resolution observations to uncover all its properties. We modified the text as follows: "They ideally should be combined with JWST continuous phase curves" (line 41).

l47: a complete orbit  more than a complete orbit

Done

l59: relative flux [...], to its neighboring pixel  This is understandable but not well formulated. For example say "Specifically, we study the relative flux in [...] by comparing it to the flux observed in neighboring pixels." or something similar.

Done

Figure 1: What is the reference frame for these spectra (planetary rest frame, stellar rest frame)?

It is in the stellar rest frame, and we change the legend accordingly.

l65: have no a priori  assume no a priori knowledge on the extent of the helium feature in WASP-121; there are (a) previous observations of WASP-121b and (b) observations of other systems, which the authors cite and use as a priori knowledge to justify their procedure, which is of course reasonable.

We agree with points a and b from the referee, but this result highlights that we should remain conservative in our approach to extract such a signal as the outflow of WASP-121b extends much farther than previous observations of this planet or other systems let suggest.

l72: Why do the authors use 37 min binning? This seems long compared to the optical transit duration.

This choice was based on the number of temporal bins, rather than their duration. Different temporal bins were tested, and they all showed similar structure. We converged on 60 bins for better readability of the data.

l76, l81: I am confused about the authors' method of co-adding. Looking at Fig. 1, the absorption in the central pixel should be a bit more than 400 ppm during the optical transit. In Fig. 2, it appears to be a lot higher (average amplitude of 2778 +/- 82 ppm). It is not clear to me how this difference comes about. Likewise, I am not entirely sure what is shown in Fig. 3. Is it the sum of the individual absorption spectra or the average spectrum during the respective periods? The authors should clarify this.

We understand where the referee's confusion comes from. In Fig. 1, the color scale was manually set from -500 to +500 ppm for visualization purposes; however, the signal maximum strength actually reaches ~2800 ppm. This is now explicitly mentioned in the legend. We attach below the same figure with no adjusted color scale as a reference for the referees. The transmission spectra in Fig. 3 are the average of the spectra obtained during the different regions described in Fig. 2.

l90: The authors use the term "phase curve" to refer to the temporal variation of the excess absorption as far as I understand. This might be confused with what is shown in Extended Fig. 1. I suggest to introduce unique terminology.

We specified that it is the helium phase curve.

l94, Table 1: blue and red pixel become left and right in Table 1.

We thank the referee for catching these typos.

Table 1: The ratios should be given with uncertainties.

Done.

l108: "meaning that the number of metastable helium atoms ahead of the planet is higher than those that are trailing"  The authors should be more precise. I think this means that the number of metastable He absorbers in front of the star is higher before the transit than after it, assuming that conditions are reasonably optically thin. As for the total number before and after, the duration of the respective periods has to be considered. I suggest the authors produce a plot showing the time evolution of the mean wavelength of the absorption component.

We replaced "number of metastable atoms" by "number density of metastable atoms". We cannot conclude on the total amount of escaping material in the two tails from this simple approach alone. The new version of Fig. 4 should better outline the density distribution.

I115: longest ****continuous**** orbital coverage  Assuming that the structure is reasonably stable in time, the continuous coverage is not actually crucial to study it although it is certainly a plus.
Done.

I121: I do not understand the significance of the L2 point here. I think the leading tail would be launched through the L1 point if such a simplification applies at all and the outflow is not approximately radial throughout a larger volume.

We apologize for the confusion; we meant the L4 point. We find it interesting, for future community-led studies, to mention that the start of the accretion is around L4, where potential Trojans may exist.

I151 and following paragraph: I think that most of the information given here would better be placed at the beginning of the paper. An upper limit on the out-of-transit non-detection of H α would be valuable information. We added more discussion on the H α line (lines 171-179). In particular, we now mention that we set a 1σ upper limit of 351ppm over 74 minutes (we choose to bin the H α light curve into 30 temporal bins). This larger uncertainty, a direct consequence of the overall lower SNR of order two, limits the possible comparison with the helium triplet signature outflow extension. Nonetheless, we further hypothesize on possible reasons for the non-detection of H α outflows.

I173 and following paragraph: It may be worth mentioning that the simulations shown by ref 29 do qualitatively show the structures reported here although they are of course based on different assumptions about the absorption.

Done.

I191: "[...] directly probing atmospheric accretion by the host."  Motion toward the star at some point is no direct probe of accretion in my opinion.

We deleted this part of the text.

I229: I don't think these have generally been overlooked. It is, however, true that most ground-based campaigns have to rely on short baselines for technical reasons.

While it is true that most studies have attempted to identify the presence of pre- or post-transit absorption, we consider that the duration of such out-of-transit absorption has been underestimated and thus overlooked. For WASP-121b, only the phase curve reveals the true nature and extent of the outflow, which could not have been done with even 15-hour observations (from -5 to +10h from mid-transit), due to the bias induced by the long plateau.

I334: "However, the amplitude of such a signature should be rather constant over its line-of-sight visibility and not vary along the phase curve as we see."  I do not understand this conclusion. I think the authors state correctly that rotational modulation can introduce a signal. The amplitude of this signal could in the extreme be similar to the stellar line EW and likely much lower. Did the authors intend to say that the time variation of a potential stellar signal would not be expected to be aligned with the planetary transit timing? If so, I agree, but the authors should clarify their statement.

We modified the sentence (lines 359-361) to make it clearer.

I336: "Furthermore, the equatorial velocity of WASP-121 is about 13.6 km s⁻¹ 13;14 , which is much lower than the velocity gradient we measure in the helium signature."  Have the authors quantified the velocity gradient? At a spectral resolution of 600 (500 km/s) this sounds like a challenging task. However, as I suggested earlier, the barycenter of absorption might tell us something about the velocity shifts and that would actually be interesting information.

Our velocity constraints are coming from the measurement of the helium triplet shape and of its ratio to the blue and red pixels, which is similar to the referee's suggestions. To match the pixel ratio, our model requires the tails to have different velocities, with the leading tail velocity to be ~+90 km/s, much larger than the equatorial velocity.

I350: While the experiences made by the authors during their fitting attempts are interesting, I suggest to focus on the results in this section. As the authors correctly point out, short baselines and the differential measurement in ground-based campaigns can lead to the described effects.

I361: "Even if some studies have reported post-transit absorption signals, it is always assumed that pre-transit data do not contain absorption signatures due to a lack of evidence until now"  As to my knowledge, this is

simply not true. People have been aware of the possibility but were often forced to make such assumptions owing to the lack of more complete phase coverage. Also pre-transit absorption was reported previously, e.g., in HAT-P-32 b (even with a "conventional" short-baseline transit campaign, see Czesla et al. 2022 and with more complete orbital sampling by Zhang et al. 2023).

We modified the text to tone down our claim (lines 387-389).

I366: "mandatory": Perhaps rather 'potential value of a reanalysis'.

Done.

I417: I think 29 should be cited here as well as this was actually discussed there for the specific case of WASP-121.

Done.

Extended Fig. 2: I think this figure is never referenced in the text.

Done.

Referee report

We thank the three referees for their positive reports, kind words, and acceptance in principle of the paper. Their feedback improved the overall quality of the paper. We have considered all their comments and answered them below in blue.

Reviewer #1 (Remarks on code availability):

A brief readme file might be helpful, otherwise the code appears self explanatory and is a helpful addition to the manuscript.

We added a README file to the Zenodo package.

Two minor points came to my attention in the revised version:

Fig. 1, caption) I suggest the authors further clarify that the scale was truncated w.r.t. that used in the other figures (as opposed to 'set') to highlight the smaller structures.

We modified the legend of figure 1 to explicitly mention truncated.

l321) I am very skeptical that gravitational darkening can introduce a strong enough asymmetry because the planetary orbit is nearly polar (leading to a symmetric signal) and although the star notably rotates, it isn't a very fast rotator like WASP-33 for example.

The referee is correct that the near-polar orbit of WASP-121 b would result in a symmetric signal from gravity darkening.

Another potential explanation would be that the planet's outflow, which is shown in this work to be asymmetric, could be optically thick up to some extent and impact the transit shape even outside the metastable helium line. Finally, an asymmetry in temperature or composition between the morning and evening limbs of WASP-121 b could also possibly introduce asymmetries in the transit shape. However, we think that a robust verification of these hypotheses would require further investigation, which is outside the scope of this work. Further investigation of this possible asymmetry will be carried out in MacDonald et al. (in prep.): a paper focusing specifically on the NIRISS SOSS transit and transmission spectrum of WASP-121 b as part of the NEAT GTO program.

Based on this, we have updated the sentence on l321 to:

We also observe an apparent asymmetry in the shape of the primary transit. Tidal distortion of the planet would result in a symmetric transit light curve, and the same is true for gravity darkening given the polar orbit of WASP-121 b. Possible scenarios for this asymmetry include uncorrected instrumental and astrophysical systematics that affect the transit shape on short timescales, or the presence of a significant temperature and molecular gradient between the morning and evening limbs of WASP-121 b. Another possibility is that the outflow of the planet, which we have found to be asymmetric, is optically thick to a certain extent and also impacts the transit shape outside of the metastable helium line. A robust verification of these hypotheses will require further modeling work.

I thank the authors for taking my comments into account.

The paragraph in question is correct, as far as I can say. As the claim of an asymmetric transit remains unsubstantiated by means of a rigorous analysis in this paper (it is "apparent" in the words of the authors), I suggest to remove the second sentence (about tidal distortion and gravitational darkening) all together and leave the reader with the possible explanations (for a possible effect). Actually, at least to my eye, this looks like a neat planetary phase curve, maybe indicative of a hot spot offset from the substellar point as has been observed in many systems before. I agree though that further modeling is required.

We thank the referee, and we have modified the text accordingly.